# Cancer-Associated Fibroblasts: Versatile Players in the Tumor Microenvironment

**DOI:** 10.3390/cancers12092652

**Published:** 2020-09-17

**Authors:** Debolina Ganguly, Raghav Chandra, John Karalis, Martha Teke, Todd Aguilera, Ravikanth Maddipati, Megan B. Wachsmann, Dario Ghersi, Giulia Siravegna, Herbert J. Zeh, Rolf Brekken, David T. Ting, Matteo Ligorio

**Affiliations:** 1Division of Surgical Oncology, Department of Surgery, UT Southwestern (University of Texas Southwestern Medical Center), Dallas, TX 75390, USA; Debolina.Ganguly@UTSouthwestern.edu (D.G.); RAGHAV.CHANDRA@phhs.org (R.C.); JOHN.KARALIS@phhs.org (J.K.); MARTHA.TEKE@phhs.org (M.T.); Herbert.Zeh@UTSouthwestern.edu (H.J.Z.III); Rolf.Brekken@UTSouthwestern.edu (R.B.); 2Department of Radiation Oncology, UT Southwestern (University of Texas Southwestern Medical Center), Dallas, TX 75390, USA; Todd.Aguilera@UTSouthwestern.edu; 3Harold C. Simmons Comprehensive Cancer Center, UT Southwestern (University of Texas Southwestern Medical Center), Dallas, TX 75390, USA; Ravikanth.Maddipati@UTSouthwestern.edu; 4Department of Internal Medicine, UT Southwestern (University of Texas Southwestern Medical Center), Dallas, TX 75390, USA; 5Department of Pathology, Veterans Affairs North Texas Health Care System, Dallas, TX 75216, USA; Megan.Wachsmann@UTSouthwestern.edu; 6College of Information Science & Technology, University of Nebraska at Omaha, Omaha, NE 68182, USA; dghersi@unomaha.edu; 7Massachusetts General Hospital Cancer Center, Harvard Medical School, Boston, MA 02114, USA; gsiravegna@mgh.harvard.edu

**Keywords:** cancer-associated fibroblasts, tumor microenvironment, heterogeneity, hallmarks of cancer, chemoresistance, immunomodulation, CAF therapeutics, clinical trials targeting CAFs

## Abstract

**Simple Summary:**

Cancer-associated fibroblasts (CAFs) are key players in the tumor microenvironment. They are responsible for potentiating growth and metastasis through versatile functions, including maintenance of the extracellular matrix, blood vessel formation, modulation of tumor metabolism, suppression of antitumor immunity, and promotion of chemotherapy resistance. As such, CAFs are associated with poor prognosis and have emerged as a focus of anticancer research. In this review, we discuss the origins of CAFs, their heterogenous subtypes and their properties. We then detail the current state of preclinical and clinical research targeting CAF activities. We believe the limited efficacy of current cancer therapeutic approaches is driven by an incomplete understanding of CAF functions and by a nonstandardized CAF classification system. Therefore, we suggest a unified CAF classification based on specific functions to develop a new class of therapies that will focus on targeting the pro-tumorigenic properties of CAFs during tumor progression.

**Abstract:**

Cancer-associated fibroblasts (CAFs) are indispensable architects of the tumor microenvironment. They perform the essential functions of extracellular matrix deposition, stromal remodeling, tumor vasculature modulation, modification of tumor metabolism, and participation in crosstalk between cancer and immune cells. In this review, we discuss our current understanding of the principal differences between normal fibroblasts and CAFs, the origin of CAFs, their functions, and ultimately, highlight the intimate connection of CAFs to virtually all of the hallmarks of cancer. We address the remarkable degree of functional diversity and phenotypic plasticity displayed by CAFs and strive to stratify CAF biology among different tumor types into practical functional groups. Finally, we summarize the status of recent and ongoing trials of CAF-directed therapies and contend that the paucity of trials resulting in Food and Drug Administration (FDA) approvals thus far is a consequence of the failure to identify targets exclusive of pro-tumorigenic CAF phenotypes that are mechanistically linked to specific CAF functions. We believe that the development of a unified CAF nomenclature, the standardization of functional assays to assess the loss-of-function of CAF properties, and the establishment of rigorous definitions of CAF subpopulations and their mechanistic functions in cancer progression will be crucial to fully realize the promise of CAF-targeted therapies.

## 1. Introduction

In 1889, pathologist Dr. Stephen Paget proposed his seminal “seed and soil” theory, highlighting the concept of tumor spread: “when a plant goes to seed, its seeds are carried in all directions; but they can only live and grow if they fall on congenial soil”. He suggested, for the first time, that the “soil” may not be altogether passive in relation to tumor pathology, progression, and metastasis [1]. After decades of focus on the cancer “seed”, a tremendous amount of energy is now focused on understanding the “congenial soil”, now referred to as the tumor microenvironment (TME). The TME, or tumor stroma, comprises multiple cell types, including fibroblasts, immune cells, endothelial cells, adipocytes, as well as extracellular matrix proteins and tumor-promoting factors (e.g., cytokines, growth factors, etc.). The complex and dynamic interactions between tumor cells and the TME have emerged, with good reason, as the focus of a skyrocketing body of research activity as we began to comprehend the fundamental contribution of the “soil” in tumor progression. To further underscore the clinical relevance of this endeavor, we point to the use of immune checkpoint inhibitors as standard of care in many cancer types. These therapies have not only added an incredibly powerful weapon to the anticancer arsenal, but also have served as proof-of-principle for developing further TME-driven therapies as a complementary avenue to therapies directly targeting cancer cells.

Cancer-associated fibroblasts (CAFs) are a critical population of cells affecting the homeostasis of the TME. Of the numerous and diverse cell types present in the stroma, CAFs are of particular interest because they are instrumental in nearly all aspects of tumor progression: extracellular matrix (ECM) remodeling, crosstalk with cancer cells, facilitating local invasion and metastasis, contributing to the regulation of peri-tumoral inflammation, and interfacing with the immune system. In light of their holistic involvement in essentially all aspects of tumor progression, attempts have been made to depotentiate CAF activity or to deplete their presence, with the hope of deriving clinical benefit. However, these endeavors have led to disappointment in the majority of cases or, even worse, the deep concern of causing harm; for example, depletion of CAFs increased the aggressiveness of pancreatic tumors [2]. How do we reconcile this finding? Recent investigation has demonstrated that CAFs are hardly a uniform population. They are, in fact, a phenotypically heterogenous mixture of cells with pro-tumorigenic and anti-tumorigenic functions [3]. As such, the paradigm of targeting CAFs has evolved and the question became: how do we identify and target pro-tumorigenic CAFs to derive clinical benefit?

However, this question has more profound implications in defining the role of CAFs and is directly entangled with these key inquiries: what are these (non-neoplastic) cells, and how shall we consider and classify them? Should we, in fact, based on the most recent data on single cell CAF heterogeneity [4,5,6,7], give up the traditional definition of CAFs based on their cell-of-origin (i.e., activated fibroblasts) and give way for a wider definition built upon their functions (i.e., ECM production, cytokine secretion, antigen presentation, etc.). A more inclusive classification of cancer-associated cells would also challenge a prevailing assumption about stroma cells that they serve as privileged cancer cell helpers during tumor progression. Such a shift in classification highlights how the distinction between cancer and noncancer cell functions is not as rigid as previously thought. More specifically, it provides a novel explanation for the epithelial-to-mesenchymal transition (EMT) phenomenon as a way for cancer cells to perform functions commonly associated with CAF under certain circumstances.

In summary, this review strives to report the recent advancements of this rapidly growing discipline to help answer these questions. Our vision for this undertaking begins with a brief synopsis of the concept of fibroblasts and how this notion has evolved over time (Figure 1), followed by a salient discussion of the cell-of-origin, CAF functions, and effects of CAFs on tumor cells (Figures 2 and 3), and concludes with an encapsulation of the contemporary state of preclinical and clinical trials (Figures 4 and 5). We attempt to identify fundamental CAF functions (i.e., factor secretion, extracellular matrix production, etc.) that can be utilized to classify CAF subpopulations and characterize them across solid tumors. Altogether, we hope this compendium sparks the new generation of scientific inquiries challenging the existing paradigms in tumor progression, facilitate the leap from bench to bedside for developing novel therapeutic avenues, and help us to understand CAFs, which are one of the most versatile and, to some extent, still obscure cell types in the TME.

## 2. History of CAFs

Fibroblasts, as a whole, were initially viewed as responders, byproducts, or passive players in the process of cancer progression. Over the years, those notions have been dispelled as they have been found to be crucial in cancer progression and even in carcinogenesis [8].

The existence of fibroblasts, as postulated in the 19th century by Virchow (1858) and Duvall (1879), was first hypothesized to explain the abundant collagen deposition observed in many human tissues (Figure 1) [9,10]. However, it was not until later that century (1889), when Paget theorized the importance of the surrounding milieu to cancer progression—“the seed and soil theory”— that the scientific community began to investigate the TME as it is currently known [1]. The concept of activated fibroblasts would take almost another 100 years to emerge (1971), when Gabianni took the first step in understanding that resident fibroblasts become activated during wound healing by acquiring some smooth muscle cell features [11]. Later that same decade (1979), activated fibroblasts, also known as myofibroblasts, were found to be important for chronic healing and hypertrophic scar formation [12], which seemed a protective place for tumor cells to seed and grow [13,14]. This discovery encouraged a deeper focus on how this framework translated into carcinogenesis, yielding the “cancer wound theory” (1986), which essentially describes cancer as “a wound that never heals” [15]. Since then, fibroblasts, and their role in the TME, have continued to gain recognition as key players in tumor progression, and as such, identifying markers of CAFs has become an increasingly fruitful undertaking.

In 1994, Schmitt-Graff and his team noted that myofibroblasts commonly expressed, upon exposure to transforming growth factor beta-1 (TGFβ-1), alpha smooth muscle actin (αSMA), a master regulator of cytoskeleton rearrangements and cellular motility [12,16,17,18,19]. This discovery opened the possibility of inhibiting this transformation by preventing TGFβ activity or by directly targeting CAF markers (e.g., αSMA), paving the way to current anti-CAF therapies. However, as an a posteriori realization, αSMA, as well as other identified CAF markers, such as vimentin, FSP-1 (fibroblast specific protein-1), and PDGFR (platelet derived growth factor receptor), are not uniquely expressed in CAFs, making therapy and biomarker discovery challenging [3,20,21,22]. The multifaceted nature of CAFs started to be recognized across different organs by Sugimoto et al. in 2006 where the authors described CAFs as a heterogenous population for the first time [21].

The appeal of CAFs as a homogenous population, and hence, a reliable anti-cancer target, was most recently challenged when preclinical and clinical studies exposed tumor restraining functions of CAFs and the unexpected progression of disease if CAFs were uniformly depleted [2,23]. These findings re-directed the field to explore the possibility of the existence of multiple CAF subpopulations, yielding either a pro- or anti-tumorigenic phenotype, evidence of which was first shown in pancreatic cancer by Ohlund et al. in 2017 [4] from David Tuveson’s lab and confirmed by other researchers in other tumor types [5,6,7,24,25].

This journey to understanding CAFs, which began almost 200 years ago, is still underway and livelier than ever. Despite the breadth of knowledge accumulated thus far, the detailed mechanisms of actions and functions of this versatile cell type in carcinogenesis and tumor progression have yet to be fully understood.

## 3. Definition and Properties of CAFs

### 3.1. Comparison of Normal Activated Fibroblasts (NAFs), Fibrosis-Associated Fibroblasts (FAFs), and CAFs

By definition, CAFs comprise all fibroblasts within a tumor mass, which are thought to interplay tightly with cancer cells. Even though CAFs can have multiple cells-of-origin (Figure 2A, and section below), resident fibroblasts are considered to be the most important source of CAFs. [4,26,27]. It is worth noting that the activation of resident fibroblasts is not a specific event to cancer but also occurs in physiologic (e.g., wound healing, acute organ repair, etc.) and non-malignant pathologic conditions (e.g., chronic infection, organ fibrosis, and autoimmune disease). In non-malignant conditions, resident fibroblasts are activated by external stimuli and become normal activated fibroblasts (NAFs) in the case of acute wound healing, or fibrosis associated fibroblasts (FAFs) in response to chronic insults [17]. The quality of the insult, and more importantly, its duration determines the fate of this activation. While NAF is a transient state, which usually ends with the cessation of the insult, scar deposition, and wound closure, FAF is a stable and active phenotype resulting in functions that are similar to CAFs, including tissue remodeling and immune modulation [17]. Corroborating this notion, a recent article comparing fibroblasts from patients with malignant and non-malignant pancreatic diseases, such as pancreatic ductal adenocarcinoma (PDAC), ampullary carcinoma and chronic pancreatitis (CP) and matched normal pancreatic tissue, found that fibroblasts from PDACs share the most similarities with activated fibroblasts derived from CP [28]. Highlighting similarities and differences between activated fibroblasts in cancer and in organ-specific chronic disease, (i.e., lung, kidney, and liver fibrosis) will be useful to gain insight into the biology of this versatile cell type during neoplastic and non-neoplastic disease.

### 3.2. Origin of CAFs

Defining the uniqueness of fibroblasts and, hence, of CAFs has been challenging [16,17,18]. In general, fibroblasts are non-immune, non-epithelial cells and non-endothelial cells that originate from the primitive mesenchyme of mesodermal cells [16,29,30]. Some fibroblasts have also been found to originate from neural crest cells which are of ectodermal origin [31]. Nonetheless, the vast majority of fibroblasts are of mesenchymal origin, similar to other cells such as chondrocytes, adipocytes, and osteoblasts [16,18]. Due to common lineages with other cell types, intrinsic inter-cellular plasticity and multiple potential sources, the cell-of-origin of CAFs remains elusive in most cancer types. Herein, we summarize the current knowledge on the origin of CAFs with a brief description of the mechanisms underlying this process, if known (Figure 2A).

#### 3.2.1. Tissue Resident Fibroblasts

Tissue resident fibroblasts, also known as quiescent fibroblasts, are one of the major sources of CAFs in tumors. They perpetrate a quiescent resting stage, being functionally inert and mitotically inactive, until they get activated in response to tissue insults or to different types of stress [26]. During tumorigenesis, resident fibroblasts engage in signaling cues from cancer cells and immune cells. These signaling molecules include CAF activators: TGFβ, receptor tyrosine kinase (RTK) ligands such as PDGF, FGF, and EGF, and pro-inflammatory molecules, such as IL-1β and IL-6, which activate resident fibroblasts through NF-κB and JAK-STAT pathways, respectively [32,33,34,35,36,37]. Parallel routes to activate CAFs include environmental stressors (e.g., reactive oxygen species, ECM stiffness, etc.) and DNA damage due to chemotherapy or radiation therapy [38,39].

#### 3.2.2. Stellate Cells

In addition to tissue resident fibroblasts, stellate cells are an important source of CAFs [4,27,40]. They are vitamin A-storing quiescent cells found in the liver and pancreas that maintain tissue homeostasis and ECM turnover [41]. Recent studies in PDAC have shown that as a result of tumorigenesis, pancreatic stellate cells (PSCs) become active, assuming CAF features [4]. This work suggests that IL-1, leukemia inhibitory factor (LIF), JAK-STAT, and TGFβ signaling are important for inducing the CAF phenotype arising from PSCs [32].

#### 3.2.3. Mesenchymal Stem Cells (MSCs)

A growing tumor, similar to wound healing, can recruit cells by secreting paracrine signals. Homing of MSCs to tumor sites involves a number of cytokines and chemokines, including CCL2, CCL5, CXCL12, secreted by cancer cells [42]. Upon recruitment, MSCs can differentiate into CAFs and in vivo studies, using labelled cells (i.e., lineage tracking techniques), have confirmed their importance in tumorigenesis as well as in metastasis in many cancer types, including breast, PDAC, and gastric cancer [43,44,45,46,47]. How these MSCs differentiate into CAFs is not clear; however, TGFβ has been suggested as an important factor in MSC differentiation [48].

#### 3.2.4. Mesothelial Cells

Mesothelial cells are a monolayer epithelium that covers body cavities in the thorax (pleura and pericardium) and in the abdomen (peritoneum). It has been thought that these cells acquire CAF-like properties by going through mesothelial-to-mesenchymal transition, aiding peritoneal seeding [49,50]. Insights into the mechanisms operant in mesothelial cell transition to CAF-like cells are not known and requires further investigation. However, recent single cell RNA sequencing (scRNA-seq) studies have shown antigen presenting CAFs (apCAFs) express some mesothelial-specific markers, suggesting their direct contribution in CAF biology [6]. Further studies are needed to confirm this fascinating hypothesis.

#### 3.2.5. Other Sources of CAFs

There are several other potential sources of CAFs whose mechanisms of differentiation into CAFs have not yet been completely elucidated. These sources might be relevant for specific cancer types. For example, adipocytes have been reported to differentiate into adipocyte derived fibroblasts (ADFs) when exposed to condition media from tumor cells [51]. In breast cancer, ADFs are important in stromal desmoplasia contributing to tumor progression [51,52]. However, the overall contribution of adipocytes to CAF populations and tumor progression in other cancer types need further exploration. Even less-defined sources of CAFs include fibrocytes, which originate from monocyte precursors and which can be recruited to the sites of injuries [53,54], local normal and tumor epithelial cells, which can undergo EMT [55,56,57,58], endothelial cells that, similar to epithelial cells, can go through an endothelial-to-mesenchymal transition (EndMT) [59], upregulating mesenchymal markers, such as FSP1, and downregulating endothelial markers, such as CD31, and pericytes, which can acquire fibroblast-like properties undergoing a pericyte-to-fibroblast transition (PFT) [60]. Given this plasticity, it is essential to gain a clearer idea on the specific contribution of the cell-of-origin of CAFs for each tumor type. Combining lineage tracing methods with single cell spatial analysis will identify the exact contribution of each cell type in tumor development and CAF heterogeneity.

### 3.3. Function of CAFS

CAFs perform highly versatile functions including tissue remodeling, mutual signaling with cancer cells and with other cells types in the TME (e.g., endothelial cells, adipocytes, etc.), and immunomodulation [16,18,61]. To achieve these complex functions, resident fibroblasts, or other CAF precursors (Figure 2A), acquire additional properties (i.e., specialized functions), in addition to proliferation and migration, such as cytoskeletal rearrangement, ECM production, cytokine secretion, and antigen-presenting capability (Figure 2B). These specialized functions are further explained in detail below.

#### 3.3.1. ECM Deposition

Similar to chronic fibrosis (e.g., liver cirrhosis), the ECM in cancer is composed of a dense collagenous network. This desmoplastic reaction is a result of copious production of ECM proteins, such as type I, III, IV, V, VII, XI, XV collagens, hyaluronic acid (HA), glycosaminoglycans, and proteoglycans [62]. Along with ECM proteins, CAFs secrete lysyl oxidases (LOX) and matrix metalloproteases (MMPs), which are matrix crosslinkers and proteases, respectively, and serve as catalytic enzymes to increase ECM stiffness and global tissue remodeling, highly contributing to tumor homeostasis and local invasion (discussed further in Section 5 of this review) [63].

#### 3.3.2. Cytoskeletal Rearrangement

Compared to resting fibroblasts, CAFs are highly contractile and motile by virtue of their cytoskeletal rearranging ability [64]. After secretion of ECM proteins, CAFs perform one of the most important tasks of tissue remodeling, stroma rearrangement. Through a combination of physical forces and secretion of matrix proteases, CAFs remodel the ECM by modulating its biomechanical properties. For example, fibronectin, which is an abundant component of the ECM, is a ligand on which CAFs can exert, through integrin-mediated focal adhesion, the mechanical forces produced by cytoskeletal contraction (i.e., contractility power) to align fibronectin fibers into parallel strands [65]. Cancer cells often exploit the contractile ability of CAFs to their advantage during local invasion, providing permissive tracks to migrate during CAF-mediated collective invasion [66].

#### 3.3.3. Factor Secretion

CAFs also influence cancer cells by engaging in autocrine and paracrine signaling via secretion of multiple growth factors: TGF-β, hepatocyte growth factor (HGF), fibroblast growth factor 5 (FGF5), LIF, growth arrest-specific protein 6 (GAS6), platelet derived growth factor (PDGF), vascular endothelial growth factor A (VEGF), stromal-derived factor-1α (SDF1), osteopontin (OPN), just to mention some, as well as a myriad of cytokines, chemokines such as IL6, IL1, CXCL2, CCL20, and extracellular vesicles (i.e., exosomes) [61,67]. These signaling are mediated by cognate receptors on cancer cells, or on other cell types, fine-tuning their functions by modulating intracellular pathways and gene expression profiles (further discussed in Section 5 of this review). In addition, the inflammatory secretome CAFs recruits immune cells, such as immunosuppressive tumor-associated macrophages (TAMs), myeloid derived suppressor cells (MDSCs), regulatory T cells (T-Regs), and neutrophils, thereby contributing to immunomodulation [61]. Recent work suggests an apolipoprotein family member, called serum amyloid A (SAA3), as master regulator of the global inflammatory CAF secretome [68]. Its ablation, in fact, produces a global downregulation of CAF cytokine and chemokines in a mouse model of PDAC [68]. 

#### 3.3.4. Antigen Presentation

Apart from immunomodulatory functions via cytokine and chemokine secretion, CAFs can also acquire antigen presenting capability, becoming functionally similar to professional antigen presenting cells (APCs), such as dendritic cells, macrophages, and B cells. It has been recently shown that this CAF subpopulation, named antigen presenting CAFs (apCAFs) [5], expresses multiple components of the MHC class II complex and has been shown in vitro to possess the ability to present antigens to T cells.

Thus, CAFs have multifaceted functions—some restraining, and some tumor promoting—which deeply impact each stage of tumor development: from initiation, to local invasion and distant metastasis as further discussed in the following sections. These distinct functions point to the importance of understanding single cell heterogeneity of CAFs given that the relative proportion of CAFs with these different activities likely affects the regional variability of the TME.

### 3.4. CAF Heterogeneity

With the advent of scRNA-seq, the complexity of CAF biology during tumor progression has increased in granularity. CAFs, which were previously perceived as a homogenous population, are now understood to be a mixture of different fibroblast phenotypes with distinct behavior [3,21]. CAF heterogeneity was initially found in genetically engineered mouse models and, then, confirmed in human tumors. This evidence inspired new scientific inquiries and brought back old unanswered questions about their biology (i.e., cell-of-origin): what are the mechanics that determine CAF fate (e.g., secretory versus myofibroblast CAFs)? Does this depend on the specific cell-of-origin? Do CAF subtypes have a preferential spatiotemporal distribution? If so, what are the molecular drivers that control this process? Can CAF subtypes differentiate into each other? Answering these and other fundamental questions in CAF biology will be crucial to comprehend the function of the TME in human tumors. To facilitate this process, we systematically review the state-of-the-art and the major advancements in this rapidly evolving discipline (Figure 2C).

#### 3.4.1. Pancreas

CAF heterogeneity has been extensively studied in PDAC. Several groups have used sequencing technologies (scRNA-seq and bulk RNA-seq) to investigate the expression profile of CAFs in mouse models and, more recently, in human PDACs (Figure 2C). The first study was performed by Tuveson’s group using a PDAC-PSC organoid system as well as in genetically engineered mouse model, KPC mice (*KrasLSL-G12D/+; Trp53LSL-R172H/+; Pdx-1-Cre*), and revealed the presence of two distinct CAF subtypes [4]: a subpopulation of CAFs near cancer cells expressing high levels of αSMA and named myofibroblast-like CAFs (myCAFs), and a subpopulation of CAFs distant form cancer cells with a high secretory expression profile named inflammatory CAFs (iCAFs). In a follow-up study, the same group showed the importance of TGFβ and IL-1, secreted by tumor cells, as major drivers of the myCAF and iCAF phenotypes, respectively.

Similar results were independently found by another group that termed myCAF-like cells “FB3 cells” and iCAF-like cells “FB1 cells” [7]. They also found FB3 CAFs expressed multiple components of the MHC class II complex. These cells were later characterized by Tuveson’s group as having antigen presenting capability and were named apCAFs. In addition, Turley’s group performed scRNA-seq on stromal cells comparing normal pancreas, early lesions and established tumors in KPP mice (*Pdx1cre/+; LSL−KrasG12D/+; p16/p19flox/flox* aka *KIC*) as well as human PDACs [6]. Based on gene expression profiling, they found two major CAF subpopulations: one with high ECM-related genes and enrichment for TGFβ-driven pathways, similar to myCAFs, and a second one with high inflammatory/immune expression profile and upregulation of IL-1 downstream pathways, similar to iCAFs. They also confirmed the presence of CAFs in late-stage human PDAC, showing that CAF heterogeneity is also evolutionarily conserved between mice and humans. Additionally, through analysis of the expression profile of normal pancreatic cells in KPP mice, they found cells co-expressing mesothelial signature along with MHC II genes, hypothesizing the mesothelial cell-of-origin for apCAFs.

These studies demonstrate a vivid and ongoing debate on CAF functions and the need for a common classification and nomenclature.

#### 3.4.2. Breast

The lack of functional studies makes the classification of CAFs even more challenging in breast cancer. Nevertheless, Costa et al. and Pelon et al. described different CAF subtypes in human breast tumors and metastatic lymph nodes [24,69]. By applying a negative enrichment strategy, CD45, EPCAM, CD31, and CD235a negative cells, followed by a positive enrichment selection by using common fibroblast markers, such as FAP, CD29, and αSMA, via fluorescence-activated cell sorting, they identified four types of CAFs [24,69]. Two of them, CAF-S2 and CAF-S3, showed a gene expression profile similar to resident fibroblasts found in normal breast tissue, while CAF-S1 and CAF-S4 were classified as myofibroblast-like cells with a pro-tumorigenic behavior, albeit through different mechanisms [24]. While reciprocal crosstalk between cancer cells and CAF-S1 promoted EMT and cell migration via CXCL12 and TGFβ, the CAF-S4 upregulated NOTCH pathway becoming more contractile, and so, facilitating cancer cell invasion [24]. This mechanistic insight was achievable by combining scRNA-Seq with functional assays. On the contrary, Bartoschek et al. focused on scRNA-seq and on histological characterization of CAFs in a mouse breast cancer genetic model (MMTV-PyMT) and found completely different CAF types: vCAF, cCAF, mCAF, and dCAFs [58]. By using tumor location, gene expression profile, and cell-of-origin, they classified vCAF and cCAF as CAFs derived from perivascular cells with cCAFs also having a proliferative phenotype, mCAFs being myofibroblast-like cells (i.e., ECM production), originating from resident fibroblasts, and dCAFs as cancer cells that underwent a complete EMT conversion. However, a recent study using a syngeneic breast cancer mouse model (4T1 mammary tumors) found the presence of myCAFs, iCAFs, and apCAFs by applying the gene expression signatures previously identified in PDAC studies [70]. This underscores the importance of common signatures and the standardization of experimental assays to validate the functional behavior of different CAF populations identified across studies.

#### 3.4.3. Others

CAF heterogeneity has not been extensively investigated in other cancer types. However, in colorectal cancer, scRNA-seq data identified two distinct CAF subpopulations, CAF-A and CAF-B [25], which were both classified as myofibroblast-like cells based on cytoskeletal and extracellular matrix remodeling genes. For most other cancer types, CAF heterogeneity is largely unexplored. For example, CAFs are present in lung cancer, prostate cancer, head and neck cancer, and cholangiocarcinoma [71,72,73,74]; however, due to the lack of in-depth functional and transcriptional characterization, it is currently very challenging to categorize them into functionally distinct states (Figure 2C).

The aforementioned studies support the evidence of substantial heterogeneity in CAFs across tumor types. However, the lack of common signatures and standardized cell-based assays to quantify CAF behavior makes reconciling CAF characterization between studies and tumor types difficult. Focusing on specialized functions, such as ECM production, cytokine secretion, antigen-presenting capability (Figure 2B), and objectively quantifying them by current assays (e.g., trans-well migration, ECM and cytokine secretion, etc.) or by implementing new ones, it might clarify for each CAF subtype its pro- or anti-tumor potential and provide insight into the complex biology of the TME.

## 4. Effects of CAFs on Cancer Cell Behavior

### 4.1. CAFs and Tumorigenesis

CAFs are pivotal in tumor progression (Figure 3) and have an intimate role in tumorigenesis itself. The notion that fibroblasts can contribute to tumorigenesis underscores the role of fibroblasts in the oncogenic pathway.

In the absence of previously formed tumor cells, loss of TGF-β responsiveness in fibroblasts were associated with increased neoplasia in adjacent benign epithelial cells in mouse models of prostate and gastric cancers [8]. Under this line of investigation, Ollila et al. demonstrated that loss of *LKB1*, the gene responsible for the Peutz-Jeghers Syndrome, in stromal fibroblasts or in mesenchymal progenitor cells resulted in polyposis in mice. The mechanism hypothesized by which these activated fibroblasts promote the hyperproliferation of their epithelial counterparts is the upregulation of the JAK-STAT3 pathway that, ultimately, leads to gastrointestinal tumors [75]. Furthermore, Maffini et al. demonstrated in a murine breast cancer model that stromal fibroblasts exposed to carcinogens promote the malignant transformation of mammary epithelial cells [76].

A parallel line of research, instead, points toward a multifactorial etiology of CAFs in tumor initiation, especially in the setting of inflammation. Erez et al. in 2010, were the first group to demonstrate the tumor-enhancing inflammatory milieu activated by CAFs in a NF-κB dependent pathway in skin neoplasia [36]. In contrast, CAF-released SDF-1, in breast cancer, binds the CXCR4 receptor on adjacent stem cells (CD44+CD24−cells), promoting their proliferation and tumor transformation [77,78]. It is therefore plausible that, along the traditional carcinoma sequence, CAFs and their precursors are a significant catalyst for malignant initiation.

Altogether, this notion speaks to an attractive perspective about a holistic role of CAFs in tumor development, and not just as an adjunct to cancer progression but as a fundamental component to tumor initiation itself.

### 4.2. CAFs and Tumor Progression

CAFs facilitate local invasion through tissue remodeling, promote tumor angiogenesis, aid the EMT of tumor cells, and foster distant metastasis (Figure 3). In this section, we will review how CAFs influence each of these processes and their main molecular mechanisms.

#### 4.2.1. Local Invasion and Angiogenesis

Cancer cells exploit CAF-mediated stromal remodeling to migrate and invade. For example, by creating a desmoplastic tumor stroma with different degrees of stiffness, CAFs facilitate focal adhesion formation while providing pro-survival signaling to cancer cells [79]. CAFs also mediate alignment of ECM proteins into parallel fibers to direct local invasion of cancer cells in a specific direction [80]. In addition, as a direct consequence of the dense desmoplastic reaction in certain tumor types, blood vessels collapse, elevating interstitial pressure [81] resulting in chronic hypoxia that promotes EMT [82,83], which fuels further migration and invasion [84]. In parallel, CAF-derived matrix metalloproteinases (MMPs) create tracks for cancer cells to invade. CAF-derived MMPs have been extensively reported to increase local invasion in multiple cancers, including breast and lung cancer [85,86,87,88]. Moreover, collective cell invasion has been reported wherein integrin-fibronectin signaling allows contact mediated migration of cancer cells in conjunction with CAFs [66,89]. Apart from modulating the ECM, CAFs can directly modify tumor vasculature. VEGF-A, produced by CAFs, induces angiogenesis, which is necessary for tumor growth [90]. CAF-derived PDGF is also important for endothelial cell migration and proliferation as well as for the CAF-derived VEGF production itself [91].

#### 4.2.2. Epithelial-to-Mesenchymal Transition (EMT) and Distant Metastasis

Multiple factors secreted by CAFs have been implicated in EMT and, hence, in local invasion. TGF-β is a widely studied CAF-derived growth factor [92], that promotes a mesenchymal phenotype in cancer cells [93,94]. Additionally, CAFs have been reported to induce EMT in lung cancer and colorectal cancer by secreting exosomes containing SNAI1 and miRNAs such as mi-R21, respectively [95,96,97,98], that activate an EMT program in cancer cells. As opposed to promoting EMT in primary tumors, CAFs recruited in the metastatic site help cancer cells undergo mesenchymal-to-epithelial transition (MET), which is essential for cancer cells to effectively colonize the target organ [99]. The transition from circulating tumor cells into metastatic initiating cells (MICs) results in reduction of the migratory phenotype (i.e., EMT) in favor of a pro-proliferative epithelial state. To facilitate the outgrowth of micro-metastasis into macro-metastasis, CAFs provide a favorable environment by producing Tenascin C (TNC), SPARC, periostin, and Tsp-1 [100,101,102,103,104]. The CAF secretome can contribute to increase the metastatic potential of cancer cells; for example, periostin stimulates Wnt signaling in breast cancer cells and promotes stemness as well colonization of distant organs [100,105].

### 4.3. CAFs and Tumor Metabolism

An intricate and dynamic metabolic relationship exists between tumor cells and CAFs (Figure 3). In 1925, Otto Warburg described what has since been termed the “Warburg Effect”—a situation in which tumor cells, in the presence of oxygen, internalize large quantities of glucose and then ferment the end product of glycolysis, pyruvate, into lactate to produce ATP, instead of producing energy via oxidative phosphorylation [106,107]. CAFs, however, are not innocent bystanders in this process. Koukourakis et al. demonstrated in colorectal adenocarcinoma that CAFs expressed high levels of proteins involved in lactate absorption, Monocarboxylate Transporter 1 (MCT1) and Monocarboxylate Transporter 2 (MCT2), as well as high levels of lactate dehydrogenase 1 (LDH1), an enzyme that catalyzes lactate oxidation. This suggests that CAFs assist neoplastic cells by buffering and reprocessing the products of Warburg metabolism [108].

A more refined understanding of the function of CAFs in tumor metabolism has led to the proposition of a new model by Pavlides et al. termed “The Reverse Warburg Effect”. The principle of the “Reverse Warburg Effect” essentially represents a metabolic symbiosis of tumor cells and CAFs. To begin this two-compartment process, tumor cells secrete hydrogen peroxide (H_2_O_2_), inducing oxidative stress upon neighboring CAFs. CAFs then begin the process of “aerobic glycolysis”— glycolysis and fermentation in the presence of adequate amounts of oxygen—resulting in increased production of energy-rich metabolic fuel, namely: pyruvate, lactate, ketone bodies, and fatty acids. These energy-rich products do not advance to undergo oxidative phosphorylation within CAF mitochondria but are instead shuttled from the CAFs to the tumor cells. The tumor cell mitochondria can then utilize these substrates for efficient ATP production via oxidative phosphorylation [109,110]. Interestingly, in triple-negative breast cancer and PDAC, increased levels of Monocarboxylate Transporter 4 (MCT4) expression, a marker for “aerobic glycolysis”, were correlated with worse prognosis [111,112]. Other than the necessity for tumor homeostasis and clinical relevance, this raises the possibility of utilizing features of CAF metabolism as a biomarker for prognostication.

CAFs are also involved in glutamine metabolism, a conditionally essential amino acid that represents a critical energy source for tumor cells. In prostate cancer, Mishra et al. show that hyperactivation of the RAS signaling pathway in CAFs via epigenetic silencing of Ras-activating Protein-like 3 (RASAL3) induces macropinocytosis and lysosomal catabolism of albumin, generating glutamine. Glutamine is then shuttled from CAFs to tumor cells, where it is converted to α-ketogutarate for entrance into the Krebs cycle [113]. Interestingly, Koochekpour et al. showed that elevated serum levels of glutamate, a product of glutaminolysis, was correlated with a higher Gleason Score (i.e., more aggressive behavior), and thereby proposed glutamate as a biomarker candidate for prostate cancer aggressiveness [114].

### 4.4. Immunomodulation

Growth, proliferation, and invasiveness of cancer cells are influenced by immune cells in the TME and CAFs (Figure 3) are important in modulating immune cell function.

#### 4.4.1. Pro-Inflammatory Milieu

Cancer is like a wound that never heals. It is associated with sustained inflammation and tissue damage which cause resident fibroblasts to become activated in response to multiple stimuli (DAMPs, IL-1β, IL-6, etc.) from the TME. Damage associated molecular patterns (DAMPs) from necrotic and dying cancer cells are sensed by CAFs causing activation of the NLRP3 inflammasome pathway (protein complexes that mediate inflammatory response to pathogen, stress, and tissue injury leading to cytokine release) and secretion of IL-1β [115]. This further perpetuates the inflammatory state of the TME. IL-1β is also an important cytokine, recently described in PDAC cancer, that drives inflammatory CAF phenotype (iCAF) phenotype in from resident pancreatic stellate cells. These iCAFs secrete a multitude of chemokines and cytokines (i.e., *CXCL1, IL-1β, IL-6, OPN*, etc.) that adds to the inflammatory milieu of the TME which further influences tumor growth, invasion, angiogenesis, metastasis, and immunosuppression [36,115,116,117].

#### 4.4.2. Immunosuppression

One of the hallmarks of cancer is immune evasion and CAFs promote immunosuppression by modulating the activity of several immune cells including CD8+ T cells, regulatory T cells, dendritic cells, neutrophils, macrophages, and myeloid derived suppressor cells in the TME.

##### Inhibition of Cytotoxic T Cells (CD8+)

Higher intratumor level of CD8+ T cells has been associated with better prognosis in various cancers, such as melanoma, colon cancer, oropharyngeal cancer, squamous cell carcinoma, and esophageal cancer [118,119]. How CAFs influence CD8+ T cells was shown in a murine model of colorectal cancer that demonstrated that CAF-derived TGF-β promoted CD8+ T cells exclusion from the TME, ultimately driving immune evasion and metastasis [120]. In another study, Lakins et al. demonstrated in vitro that lung cancer CAFs presented antigens to CD8+ cells in an MHC I-dependent manner and suppressed their cytotoxic activity via expression of FAS- Ligand and Programed Death-Ligand 2 (PD-L2) [121].

##### Upregulation of Regulatory T Cells (CD4+, FOXP3+)

Physiologically, regulatory T cells (Tregs) are one of the important regulators of the adaptive immune system, maintaining self-tolerance, preventing autoimmunity, and regulating the duration and magnitude of the inflammatory response [122]. They, in fact, broadly control the activity of T cells (i.e., non-T reg *CD4+/FOXP3−*function) and dendritic cells via cell-to-cell contact inhibition (e.g., Galectin-1 expression, LAG-3, etc.) and by secreting numerous immunosuppressive cytokines (e.g., IL-10, TGFB) [123,124,125]. Given their immunological importance, the crosstalk between CAF and Tregs is under investigation. In a breast cancer model, Costa et al. demonstrated that CAFs promoted Treg recruitment via secretion of CXCL12. In another study, Fu et al. noted breast cancer CAFs co-cultured with Tregs stopped cycling and were arrested in the G0/G1 phase, leading the authors to hypothesize an intricate cross-talk between these two cell types. Given this connection, the interaction between CAFs and Tregs merits further investigation.

##### Modulation of Dendritic Cells

Dendritic cells (DCs) are professional antigen-presenting cells (APCs) in the body that can expresses antigen via Class I and II MHC complexes [126,127]. As such, cancer cells must subvert DC activity to escape immune surveillance often through secretion of TGF-β [128]. Recently, a subpopulation of DCs has been identified with marked immunosuppressive activity, named regulatory DCs (rDCs). It has been shown in PDAC and lung cancer rDCs promote immunosuppression through impaired antigen presentation (i.e., lipid oxidation and inhibition of MHC-I complex expression), diminished expression of costimulatory signals (i.e., CD80 and CD86), secretion of immunosuppressive cytokines (i.e., IL-10, TGF-β) and metabolic derangement (i.e., tryptophan depletion with subsequent CD8+ T cell and NK dysfunction) [128,129,130,131,132,133,134]. The IL-6/JAK/STAT3 pathway appears to be critical in this immunosuppressive process as in an in vitro hepatocellular carcinoma model IL-6 secreted by CAFs activates JAK/STAT3 pathway in rDCs, reducing their co-stimulatory signaling (e.g., CD80, CD86, etc.) and increasing their immunosuppressive cytokine production (e.g., IL-10, TGF-β, etc.) [131].

##### Polarization of Tumor-Associated Macrophages

Tumor-associated macrophages (TAMs) are an important component of the TME. While TAMs can phenotypically exist anywhere between anti-tumorigenic M1 and pro-tumorigenic M2 phenotype, TAMs are considered more to be M2-like. Many studies have shown that M2-TAMs exert their pro-tumorigenic potential by enhancing immune tolerance, angiogenesis, tumor growth, local invasion, and metastasis [135,136,137]. As such, high levels of M2-TAMs are associated with worsened prognosis in multiple tumor types including esophageal, breast, pancreatic, ovarian, and gastric cancers [135,136,138,139]. CAFs contribute to the biology of TAMs. CAFs promote the recruitment of monocytes (i.e., macrophage precursors) into the TME and mediate their polarization toward the M2 state. In a colorectal cancer model, Zhang et al. demonstrated that through IL-6 secretion, CAFs enhanced cancer cells’ expression of VCAM-1, a cell adhesion molecule frequently used by endothelial cells to recruit lymphocytes and monocytes to the site of injury [136]. CAFs also contribute to M2 polarization by secreting multiple cytokines and stimuli including macrophage-colony stimulating factor, reactive oxygen species, IL-6, IL-8, CCL2, and CXCL6 as seen in multiple cancer model systems (e.g., PDAC, colorectal and esophageal squamous cell carcinoma) [136,138,140].

##### Suppression of Natural Killer Cells

Natural killer (NK) cells are members of the innate immune system, and they mediate their immune surveillance activity via secretion of cytolytic enzymes (i.e., perforins, granzymes). It has been shown that CAFs can suppress NK function in multiple ways, both directly and indirectly, facilitating immune evasion of cancer cells. In the aforementioned study by Zhang et al., in which CAFs increased TAM recruitment, the authors also noted that TAMs suppressed NK cytotoxic activity [136]. CAFs may also directly enhance the ability of tumor cells to evade NK surveillance. In an in vitro melanoma model, CAFs produced MMPs, which resulted in the abrogation (through protein cleavage) of the activity of two cell surface proteins, MICA-A and MICA-B (MHC Class-I polypeptide-related sequence A/B), which are inducers of NK activation [141]. Furthermore, Huang et al. demonstrated that TGF-β-induced secretion of IL-6 from CAFs inhibited NK cell activity in in vitro and in vivo murine PDAC models [142]. CAFs may also directly suppress NK cytotoxicity, as demonstrated by their ability to reduce NK secretion of perforins and granzymes as shown in an in vitro colorectal cancer model [143].

##### Recruitment of Neutrophils

Polymorphonuclear leukocytes (PMNs) are among the first cells recruited in response to an injury or tissue damage. While their role in cancer is poorly understood, they may be involved in maintenance of tumor inflammation and in disease progression, as seen in breast, lung, and pancreatic cancer [134,144,145,146]. The effect of CAFs on PMN biology has been recently postulated. Cheng et al. showed in an in vitro HCC model that CAFs can recruit PNMs via CXCL12 secretion and activate them by secreting IL-6. As a result of IL-6 secretion with the following activation of the JAK/STAT3 pathway, PMNs upregulate the expression of PD-L1 (programmed death-ligand 1), which in turn, induces T cell death by binding to PD-1. Another hypothesized mechanism on how CAFs may influence PMN function is through the secretion of amyloid-β, which in turn induces the formation of neutrophil extracellular traps or NETs (extracellular chromatin network containing protein granules of myeloperoxidase, elastase, cathepsin, etc.) creating a potentially pro-tumorigenic microenvironment [147].

##### Promoting Infiltration of Myeloid-Derived Suppressor Cells

Myeloid-derived suppressor cells (MDSCs) are involved in multiple aspects of immunomodulation, including T-cell suppression [148,149,150]. The interplay between CAFs and MDSCs remains largely obscure. However, recently, Xiang et al. demonstrated in squamous cell lung cancer that CAFs recruit monocytes via secretion of CCL2 and induce an MDSC-like phenotype [151]. Similar studies by Deng et al. showed in an HCC in vitro model that CAFs can directly differentiate monocytes, upon their recruitment (via CXCL12) into MDSC cells in a JAK/STAT3-dependent manner via IL-6 secretion [131]. In these two model systems, monocyte derived MDSC acted as potent immune suppressors by directly inhibiting T CD8+ activity.

### 4.5. CAFs and Mechanisms of Chemoresistance

The lethality of many cancers can be attributed to the insensitivity of tumor cells to chemotherapy and targeted therapy. Evidence has accumulated that this is due, in part, to the actions of CAFs through two broad mechanisms: (1) secretion of factors (e.g., cytokines and extracellular vesicles) that increase the chemorefractory potential of cancer cells and (2) impairment of drug delivery to the tumor mass.

#### 4.5.1. CAF Secreted Factors

In non-small cell lung cancer (NSCLC), IL-6 induces EMT, resulting in cisplatin resistance [152], while melanoma circumvents BRAF inhibition by reactivating downstream pathways (i.e., MAPK, PI3K-AKT) through c-Met activation via CAF-derived HGF (Hepatocyte Growth Factor) [153]. Additionally, Su et al. demonstrated in NSCLC and breast cancer that a population of CAFs secrete IL-6 and IL-8, sustaining cancer stem cell renewal and resistance to chemotherapy [154]. Another study by Brechbuhl et al. suggested that breast cancer cells exhibited tamoxifen resistance when co-cultured with a specific subpopulation of CAFs, CD146-negative CAFs, and that this phenotype can be fully rescued by exposing the tumor cells to conditioned media from CD146-positive CAFs [155].

A parallel mechanism of CAF-mediated chemoresistance is via exosome secretion. CAF can transfer intracellular elements, such as proteins, mRNAs, long-non-coding RNAs (lncRNA), and miRNAs, to cancer cells, directly modifying their behavior. Richards et al. showed that in PDAC, CAFs, upon gemcitabine exposure, release exosomes, making cancer cells resistant to chemotherapy through an EMT-mediated mechanism by delivering Snail, and one of its targets (miR-146a) [156]. Similarly, Ren et al. showed that colorectal cancer cells exhibited oxaliplatin chemoresistance, through Wnt/β-catenin activation, via lncRNAs (lncRNAH19) transferred by exosomes [157]. Collectively, these findings suggest that inhibiting exosome-mediated transfer has potential as a cancer therapeutic.

#### 4.5.2. Impairment of Drug Delivery

Leung et al. showed that CAF-induced upregulation of LPP (Lipoma Preferred Partner) in microvascular endothelial cells increased the leakiness of vessels destined to feed tumor cells, resulting in reduced delivery of drugs administered intravenously [158]. Another mechanism by which CAFs may greatly contribute to tumor chemoresistance is through a reduction of vascular density and tumor microvasculature compression as a consequence of the dense desmoplastic reaction seen in many solid tumors (e.g., PDACs, breast cancer, etc.). Olive et al. and Hingorani et al. independently demonstrated, through two distinct mechanisms of action, that reducing tumor stroma density and, hence, its interstitial pressure significantly improved drug delivery [159,160].

The above mechanisms represent evidence that CAFs also contribute to tumor progression by mediating therapy resistance. A better understanding of the molecular mechanisms underlying CAF-driven drug resistance may elucidate novel therapeutic strategies to treat cancer patients.

## 5. Clinical Impact of Targeting CAF Activity

CAFs promote virtually every aspect of the hallmarks of cancer. Expectedly, the presence of CAFs in cancer specimens from patients has been associated with a poor prognosis in multiple cancers [6,161,162]. For example, in gastric cancer, a high tumor stroma ratio is associated with a worse prognosis, and in oral squamous cell carcinoma, high CAF density is associated with an increased risk of mortality. Furthermore, Dominguez et al. demonstrated that elevated expression of LLRC15+ myCAFs is associated with poor response to anti-PD-L1 immunotherapy in multiple cancer models, including for PDAC, bladder cancer, renal cell carcinoma, and NSCLC [6]. For this reason, CAFs are ideal candidates for targeted therapies in multiple solid tumor models. As would be expected, recognition of the promise of CAF-directed therapy has led to an upsurge of research activity (Figure 4) including dozens of clinical trials (Figure 5). Potential strategies for CAF-targeted therapy can be broadly separated into two categories: inhibition of pro-tumorigenic functions and promotion of antitumorigenic CAF functions, and research thus far has been dominated by the former. Below, we categorize and detail the primary avenues currently underway to target CAF activity, highlighting the most contemporary advances.

### 5.1. Antagonizing CAF Activity

#### 5.1.1. Inhibition of the CAF Secretome

##### TGF-β

TGF-β is a multifunctional cytokine that has a broad impact on tumor tissue homeostasis in many solid cancers. Interest in TGF-β inhibition is supported by favorable preclinical findings in multiple tumor models [163,164,165,166,167,168,169,170,171,172]. Clinically, a phase Ib/II trial of galunisertib (a small molecule inhibitor of the type I TGF-β receptor) plus gemcitabine improved overall survival versus gemcitabine alone in patients with unresectable PDAC [173]. Additionally, a phase II trial of galunisertib for treatment of HCC is underway (NCT01246986). Phase I trial data of another agent, fresolimumab, an anti-TGF-β monoclonal antibody, preliminarily suggested antitumor activity in melanoma and renal cell carcinoma [174]. Moreover, a phase II trial of fresolimumab plus radiotherapy improved survival of patients with metastatic breast cancer [175]. Another approach involves TGF-β gene silencing with trabedersen (an antisense oligonucleotide specific for TGF-β2 mRNA) which reduced tumor growth in a PDAC mouse model [176] and was suggested to have possible efficacy in a phase I trial [177]. Yet another agent under investigation is M7824, a trap fusion protein designed to simultaneously block PD-L1 and TGF-β, suppressed tumor growth in mouse models of multiple tumors [178]. Phase 1 trial data showed signs of possible efficacy [179] and numerous phase II and phase III trials are underway (Figure 4).

##### Fibroblast Growth Factor

Fibroblast growth factor receptor (FGFR) signaling has been implicated in carcinogenesis [180] which has yielded interest in exploiting it as a therapeutic target. Erdafitinib, a tyrosine kinase inhibitor of FGFR, demonstrated antitumor activity in locally advanced and metastatic urothelial carcinoma, and has received FDA approval [181]. Erdafitinib also suggested some efficacy in a phase I study for treatment of cholangiocarcinoma [182] and is under investigation for treatment of other solid tumors (NCT03238196, NCT02421185, NCT03210714, NCT03999515). Another FGFR inhibitor, AZD4547, suggested antitumor activity in breast cancer patients in a phase II trial [183]; however, phase II trials failed to show clinical benefit in other cancers including mesothelioma [184], squamous cell lung cancer [185], and gastric cancer [186]. Additionally, fisogatinib, another FGFR inhibitor, elicited a modest clinical response in a phase I trial in HCC. [187], and a phase II trial is underway (NCT04194801).

##### CCN2

CCN2 (Centralized Communication Network *2*) is a matricellular protein overexpressed by CAFs that acts as an adaptor between the cell surface and ECM [188]. Hutchenreuther et al. found that deletion of CCN2 in a mouse model resulted in reduced CAF activation; moreover, the degree of CCN2 expression in melanoma negatively correlated with patient survival [189]. Pamrevlumab, a monoclonal antibody targeting CCN2, suggested efficacy in a phase I trial in PDAC [190]. Phase II (NCT02210559) and Phase III (NCT03941093) trials are underway for treatment of unresectable PDAC.

##### SHH

SHH inhibitors have been extensively investigated for the treatment of multiple cancers with widely varied clinical responses, ranging from FDA-approved therapy in advanced basal cell carcinoma to a detrimental effect in PDAC patients. SHH inhibitors target the canonical SHH pathway by preventing the activation of smoothened (SMO) [191]. Vismodegib and sonidegib, inhibitors of the SHH pathway, were demonstrated to be efficacious in the treatment of basal cell carcinoma, and are FDA-approved therapies [192]. In a phase II trial, vismodegib has suggested efficacy in some pediatric patients with medulloblastoma [193]. Conversely, no clinical benefit was observed in phase II trials in other tumors including: PDAC [194,195], colorectal cancer [196], gastric and gastroesophageal junction carcinoma [197], small cell lung cancer [198], and ovarian cancer [199]. On the contrary, sonidegib showed possible efficacy in triple negative breast cancer patients in a phase I trial [200]. Of note, a phase I trial of sonidegib in combination with pembrolizumab (an immune checkpoint inhibitor) in patients with advanced solid tumors is underway (NCT04007744).

#### 5.1.2. Reverting CAFs to a Quiescent State

##### Vitamin D Receptor Agonism

Sherman et al. showed that activation of the vitamin D receptor (VDR), a transcriptional suppressor of the activated CAF state, led to enhanced drug delivery to mouse PDAC models [201]. The VDR agonist in combination with gemcitabine improved survival in these mouse models that has led to a series of clinical trials. These trials include a phase II placebo controlled randomized trial of the VDR agonist paricalcitol plus gemcitabine and nab-paclitaxel in metastatic PDAC (NCT03520790), a phase II pilot of paricalcitol, gemcitabine, nab-paclitaxel, and cisplatin (NCT03415854), and a pilot study of perioperative paricalcitol, gemcitabine, nab-paclitaxel, and nivolumab in resectable PDAC (NCT03519308).

##### Vitamin A Receptor Agonism

The vitamin A analogue, ATRA (all-trans retinoic acid), may reprise the quiescent state of CAFs via RAR-β (retinoic acid receptor-β)-mediated downregulation of actomyosin contractility, thereby reducing CAF-mediated matrix remodeling and migration [202]. Phase II trials are underway in PDAC (NCT04241276), breast cancer (NCT04113863), and prostate cancer (NCT03572387).

### 5.2. Inhibition of ECM Production and Stroma Remodeling

#### 5.2.1. Hyaluronic Acid

Reduction of HA deposition by CAFs has been investigated with the intent of reducing desmoplasia, thereby alleviating compression of the tumor microvasculature and enhancing delivery of chemotherapy. In the preclinical setting, minnelide reduced the desmoplastic stroma in PDAC [203]; a phase II trial is underway (NCT03117920). Another agent, PEGPH20, a hyaluronidase, suggested favorable outcomes in a phase II trial [159]; however, the subsequent phase III trial was terminated due to lack of efficacy (NCT02715804).

#### 5.2.2. Collagen

Inhibition of CAF-mediated collagen secretion and cross-linking represents another avenue of active research. Losartan is an indirect inhibitor of Type I collagen synthesis and has shown efficacy in preclinical models of colorectal cancer [204], ovarian cancer [205], breast cancer [206], and lung cancer [206]. A phase II trial of losartan administered with FOLFIRINOX and radiation in the neoadjuvant setting downstaged locally advanced PDACs in 61% of cases [207]. An additional phase II trial of losartan, in combination with nivolumab (immune checkpoint inhibitor) plus FOLFIRINOX and radiation, is underway (NCT03563248). Another approach is the inhibition of collagen cross-linking via inhibition of lysyl oxidase (LOXL2) by the monoclonal antibody, simtuzumab. Phase II trials, however, failed to show benefit in PDAC [208] and colorectal cancer [209].

#### 5.2.3. Photothermal Therapy

Photothermal therapy represents a novel mechanism for inducing stromal remodeling that is characterized by generation of controlled hyperthermia in the tumor via implantation of nanoparticles which can be remotely activated by a laser or magnetic field. In a preclinical cholangiocarcinoma model, Nicolás-Boluda et al. reported a significant reduction of tumor stiffness following iron oxide-gold nanoparticle-mediated photothermal therapy, which was due to preferential nanoparticle uptake by CAFs, resulting in CAF depletion and reduced desmoplasia [210].

### 5.3. CAF-Specific Immunotherapy

CAF-specific immunotherapy is an emerging treatment direction. One potential target is the CXCL12/CXCR4 (C-X-C ligand 12/C-X-C chemokine receptor 4) axis, which may be involved in immune evasion. Feig et al. showed rapid accumulation of T cells in tumors after treatment with a CXCR4 antagonist in a preclinical PDAC model [211]. Balixafortide, a CXCR4 antagonist, has suggested possible efficacy for treatment of metastatic breast cancer in a phase I trial [212].

Another strategy under investigation is vaccination against proteins expressed by CAFs. Loeffler et al. report that administration of a DNA vaccine inducing immunity against FAP (fibroblast activation protein), a protein overexpressed by CAFs, resulted in increased T-cell mediated killing of CAFs, reduced tumor growth, and improved survival in a murine model of colon and breast cancer [213]. Another vaccine under investigation is Belagenpumatucel-L, a vaccine composed of tumor cells transfected with a TGF- β2 antisense vector; this has been hypothesized to work by increasing tumor antigen recognition and inhibiting the immunosuppressive function of TGF-β2. Nemunaitis et al. reported favorable results in a phase II trial in NSCLC [214]; however, a phase III study failed to meet its primary endpoint [215] (Figure 4).

## 6. Conclusions

In this review, we detailed our current understanding of CAF biology and shed light on the leading-edge advances related to CAFs. We outlined the profound impact that CAFs have on the cellular composition, structure, and function of the TME, highlighting the important role CAFs have on multiple hallmarks of cancer. An improved contemporary understanding of the versatile nature of CAFs, has unsurprisingly, led to a flood of research activity with the hopes of targeting CAFs therapeutically. The great enthusiasm for considering CAFs an appealing anti-tumor target, as reflected by the exponential growth of CAF-specific trials (*n* = 175), has only been surpassed by the disappointment in their failure to demonstrate efficacy with only three compounds having achieved FDA approval over the last 15 years (Figure 4 and Figure 5). Modern omics technologies have helped us realize that CAFs have multiple functions: some pro-tumorigenic and some anti-tumorigenic. This insight was immediately translated into the paradigm shift of targeting distinct CAF subtypes (i.e., pro-tumorigenic CAFs), instead of considering CAFs as a homogenous population. However, learning from the mistakes of the past will be mandatory to avoid a multiplication of future negative trials (Figure 4 and Figure 5). To this end, a uniform nomenclature and the standardization of functional assays to test the loss-of-function of specific CAF properties are, in our opinion, the most urgent issues that need to be addressed, along with the identification of non-promiscuous CAF targets (i.e., targets that only inhibit pro-tumorigenic functions). In parallel, a rigorous definition of CAF subpopulations, their mechanistic functions in tumor progression, and the molecular mechanisms that drive CAF heterogeneity (e.g., epigenetic factors, cell of origin, etc.) in physiologic (i.e., wound healing) and in pathologic conditions (i.e., chronic inflammation in non-neoplastic diseases), are additional areas for promising research. Altogether, CAFs are dynamic players in the TME. There are multiple challenges that the next generation of basic and physician scientists will need to face in order to translate groundbreaking discoveries in CAF biology into modern therapeutic options for cancer patients.

## Figures and Tables

**Figure 1 cancers-12-02652-f001:**
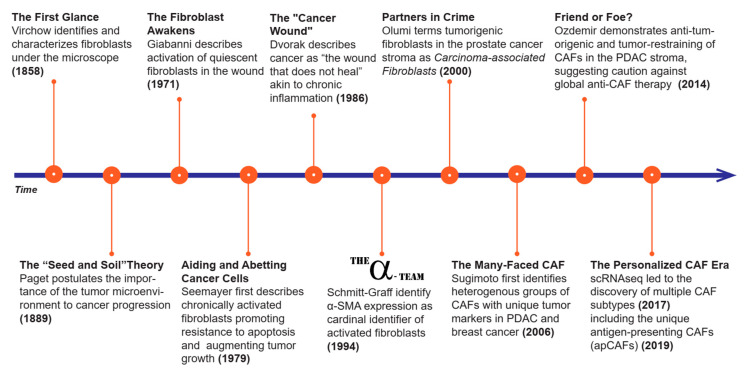
History of cancer associated fibroblasts (CAFs). This figure represents a timeline of CAF related discoveries. Here we highlight the major landmarks in CAF research that helped advance this field.

**Figure 2 cancers-12-02652-f002:**
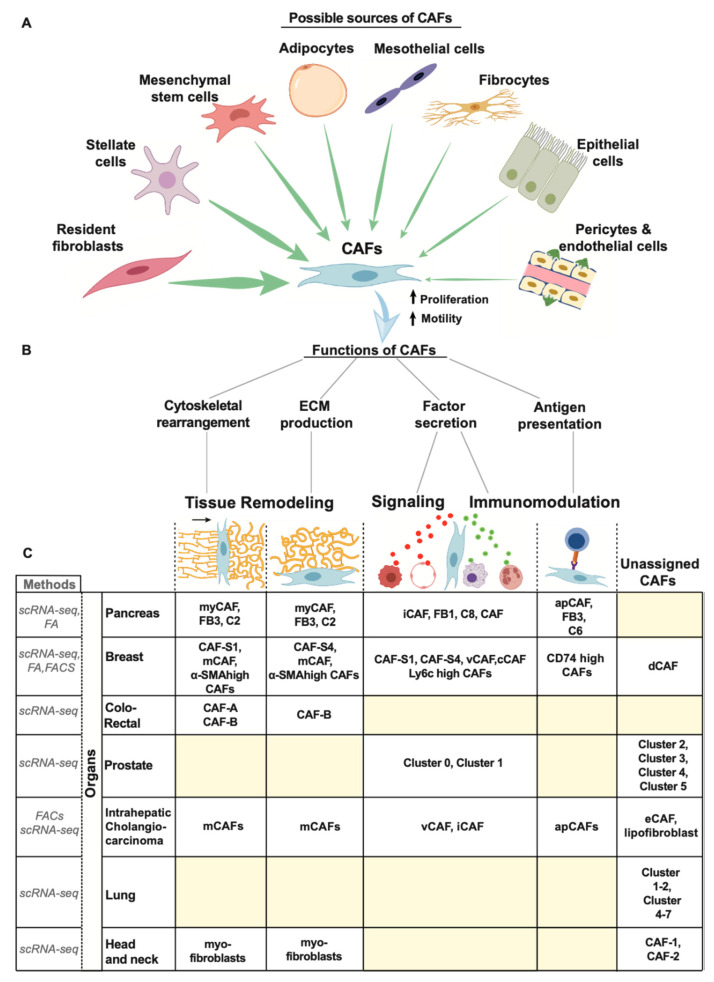
Origin, function, and heterogeneity of CAFs. This is a schematic representation of the (**A**) different cell origins of CAFs, (**B**) various functions of CAFs, and (**C**) heterogeneity of CAFs as found in different organs based on current studies. The yellow squares in (**C**) indicates yet un-characterized CAF subtype for that particular tumor. Abbreviations: FA, functional assay; FACs fluorescence-associated cell sorting; scRNA-seq, single-cell RNA sequencing.

**Figure 3 cancers-12-02652-f003:**
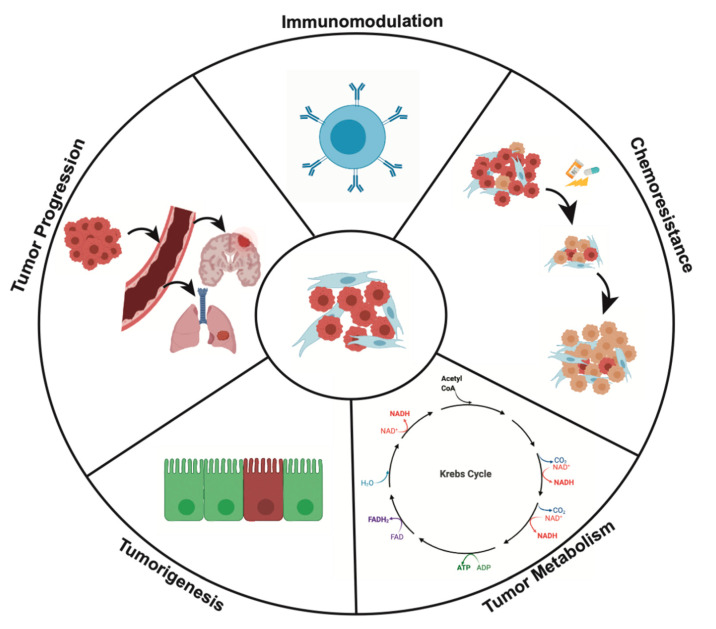
CAFs’ effect on the hallmarks of cancer. This diagram highlights the processes in tumor development that CAFs play a role in especially immunomodulation, tumor progression, tumor metabolism, tumorigenesis, and chemoresistance.

**Figure 4 cancers-12-02652-f004:**
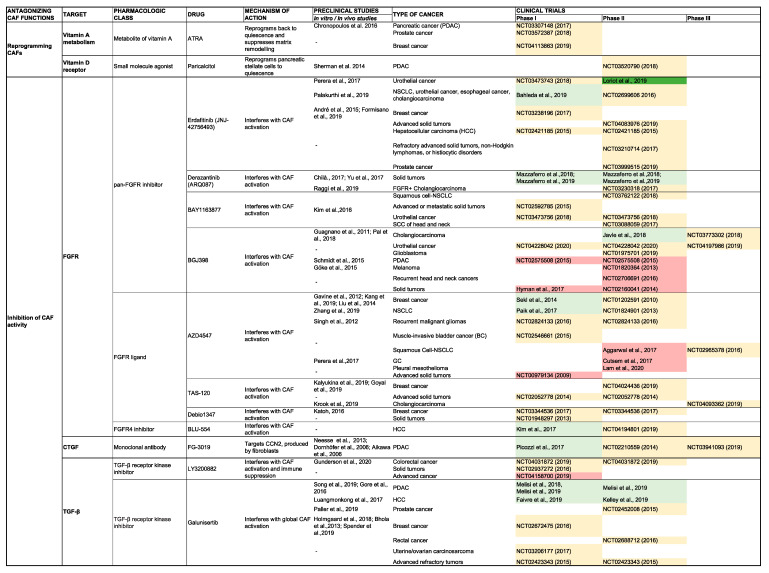
Clinical trials targeting CAFs or CAFs’ functions. This lists most of the clinical trials targeting CAF biology. The results of these clinical trials are highlighted in light green (published clinical trials with positive results), yellow (ongoing or unpublished clinical trials), red (failed/discontinued), dark green (trials producing an FDA approval). References for all these clinical studies are from https://clinicaltrials.gov/. Abbreviations: BCC, basal cell carcinoma; CAF, cancer-associated fibroblast; FGFR, fibroblast growth factor receptor; HCC, hepatocellular carcinoma; NSCLC, non-small cell lung cancer; PDAC, pancreatic ductal adenocarcinoma; SCC, squamous cell carcinoma; SCLC, small cell lung cancer.

**Figure 5 cancers-12-02652-f005:**
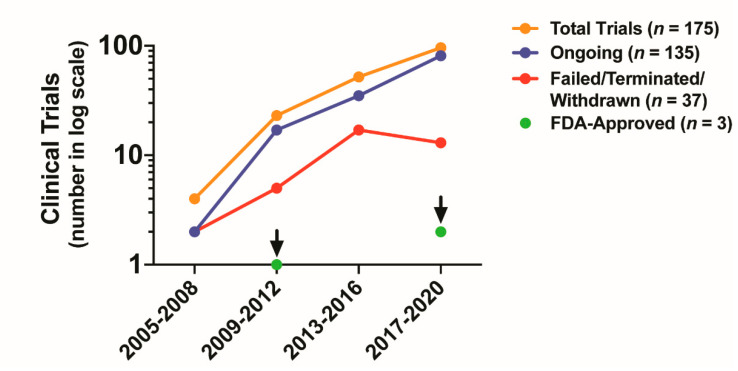
Account of clinical trials targeting CAFs or CAFs’ functions. The graph represents number of clinical trials (non-cumulative) targeting CAFs over time. Total number of clinical trials are indicated in orange (*n* = 175), ongoing clinical trials in blue (*n* = 135), failed or terminated or withdrawn trials in red (*n* = 37), and FDA-approved drugs in green. Arrows indicate the time period when the trial of the FDA-approved drug was published (*n* = 3)

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
