# Peer review of "Cancer-Associated Fibroblasts: Versatile Players in the Tumor Microenvironment"

_cancers, 2020, doi:10.3390/cancers12092652_

Round 1

Reviewer 1 Report

The manuscript by Ganguly and colleagues reports, in a nicely presented format, some of the recent advancements in the field of cancer-associated fibroblasts (CAFs). The manuscript starts with the discussion of the first reports regarding CAFs in the seventies and ends on recent clinical trials aiming at limiting CAFs’ functions in cancer patients.

As a non-expert in this field of research, I found the review very interesting and easy to read. I thus believe it will be of interest for a vast variety of readers.

That said, authors often indicate that CAFs are diverse (in term of origins and functions), and it is unclear to me how this information is taken into account in dissecting their function (axis 4) or in drug development and clinical trials (axis 5).

It would thus be nice if authors could speculate/discuss a little bit more on:

- How do they see the field unfolding both at the technical and conceptual levels in the next years to understand CAFs functions?

- Whether resistance to “anti-CAF treatments” are expected (as for cancer cells) ? Whether common side effects are also anticipated? Why is it interesting to develop such “anti-CAFs” compounds (did I miss this info?)?

- Discuss more the anti-tumorigenic functions of CAFs (mentioned on lane 66) but not really described in the subsequent sections. Is it possible to boost these functions as an anti-cancer strategy?

- Is there some clinical data showing a correlation between the proportion of CAFs (or their nature) in a tumor and the behavior of that particular tumor or the survival of cancer patients?

- How different are human and murine CAFs? Can it impact preclinical assays?

- In the same line of questioning : It is believed that patient-derived xenograft models retain features of the microenvironment? Are findings on CAFs challenging this information?

Minor:

Some important studies could be cited such as CAFs in breast cancer (Costa et al., 2018, Cancer Cell).

Lane spacing is not homogenous throughout the manuscript (just a comment).

Some acronyms are not explained: scRNA-seq (lane 275), PDAC-PSC (lane 291), FACS (lane 318)… (just a comment).

Reviewer 2 Report

In this interesting review, Ganguly et al., give a comprehensive overview of the multiple functions performed by cancer-associated fibroblasts (CAF) during tumor initiation, progression, and therapeutic treatment. They summarize the main cells of origin from which CAF derive as well as the different subtypes of CAF populations that have been described so far in the tumor. Then they explain the different ways in which CAF can alter other players inside the tumor, including tumor epithelial cells, as well as other stromal components. Finally, they give an overview of the mechanisms related to therapy resistance mediated by CAFs, as well as the new therapies directed specifically against CAF. In general, the manuscript is well written and organized, and it is very enjoyable to read. Figures are well illustrated and complement the text well. Just a couple of issues should be addressed before the manuscript is accepted for publication in Cancers.

  • Section 3.4 “CAF heterogeneity” is one of the most interesting ones and one of the more actives in the field of CAF research. A recent paper by Zhang et al (PMID: 32505533) describes 6 differents populations of CAF in cholangiocarcinoma by sc-RNA-seq, some of which match with some of the populations described in the review. This paper should be incorporated to the review in order to give strength to the point of creating a consensus classification of CAF subpopulations, to which the reviewer totally agrees. Perhaps, authors should also check for other recently published papers in this line.
  • Similarly, section 5 “Clinical impact of targeting CAF activity” is one of the most interesting ones. In spite of all the subsections describing different methods to target CAF, one failed to find a subsection dedicated to targeting CAF by physical means, such as the one described by Nicolas-Boluda et al (PMID: 32338871), also in cholangiocarcinoma, that described a photothermal approach to deplete CAF and therefore normalize tumor stiffness. Such a section should be incorporated.
  • There are some minor defects in format in some of the paragraphs. Line spacing is different among some paragraphs. Some examples: lines 468-474, lines570-577, lines623-629 and lines 675-678,
